# THE (UN)RELIABILITY OF SALIENCY METHODS

## ABSTRACT

Saliency methods aim to explain the predictions of deep neural networks. These methods lack reliability when the explanation is sensitive to factors that do not contribute to the model prediction. We use a simple and common pre-processing step —adding a mean shift to the input data— to show that a transformation with no effect on the model can cause numerous methods to incorrectly attribute. We define input invariance as the requirement that a saliency method mirror the sensitivity of the model with respect to transformations of the input. We show, through several examples, that saliency methods that do not satisfy a input invariance property are unreliable and can lead to misleading and inaccurate attribution.

## 1 INTRODUCTION

While considerable research has focused on discerning the decision process of neural networks (Baehrens et al., 2010; Simonyan & Zisserman, 2015; Haufe et al., 2014; Zeiler & Fergus, 2014; Springenberg et al., 2015; Bach et al., 2015; Yosinski et al., 2015; Nguyen et al., 2016; Montavon et al., 2017; Zintgraf et al., 2017; Sundararajan et al., 2017; Smilkov et al., 2017; Kindermans et al., 2017), there remains a trade-off between model complexity and interpretability. Research to address this tension is urgently needed; reliable explanations build trust with users, helps identify points of model failure, and removes barriers to entry for the deployment of deep neural networks in domains high stakes like health care and others.

In deep models, data representation is delegated to the model. We cannot generally say in an informative way what led to a model prediction. Instead, saliency methods aim to infer insights about the $f(x)$ learnt by the model by ranking the explanatory power of constituent inputs. While unified in purpose, these methods are surprisingly divergent and non-overlapping in outcome. Evaluating the reliability of these methods is complicated by a lack of ground truth, as ground truth would depend upon full transparency into how a model arrives at a decision — the very problem we are trying to solve for in the first place (Sundararajan et al., 2017; Kindermans et al., 2017).

Given the need for a quantitative method of comparison, several properties such as completeness (Bach et al., 2015; Sundararajan et al., 2017), implementation invariance and sensitivity (Sundararajan et al., 2017) have been articulated as desirable to ensure that saliency methods are reliable. Implementation invariance, proposed as an axiom for attribution methods by Sundararajan et al. (2017), is the requirement that functionally equivalent networks (models with different architectures but equal outputs all inputs), always attribute in an identical way.

This work posits that a second invariance axiom, which we term *input invariance*, needs to be satisfied to ensure reliable interpretation of input contribution to the model prediction. Input invariance requires that the saliency method mirror the sensitivity of the model with respect to transformations of the input. We demonstrate that numerous methods do not satisfy input invariance using a simple transformation –mean shifts of the input– that does not affect model prediction or weights. We limit our treatment of input invariance to showing that there exist cases where this property is not satisfied and welcome future research on a broader treatment of this topic.

In this work we:

- introduce the axiom *input invariance* and demonstrate that certain saliency methods do not satisfy this property when considering a simple mean shift in the input. (See Fig. 3).
- show that when input invariance is missing, the saliency method becomes unreliable and misleading.

## Attribution By Reference Point

Figure 1: Integrated gradients and Deep Taylor Decomposition determine input attribution relative to a chosen reference point. This choice determines the vantage point for all subsequent attribution. Using two example reference points for each method we demonstrate that changing the reference causes the attribution to diverge. The attributions are visualized by multiplying them with the input image as is done in the IG paper[1] (Sundararajan et al., 2017). Visualisations were made on ImageNet (Russakovsky et al., 2015) and the VGG16 architecture (Simonyan & Zisserman, 2015).

- demonstrate that "reference point" methods—Integrated gradients and the Deep Taylor Decomposition—have diverging attribution and input invariance breaking points that depends upon the choice of reference (Fig. 1).

In **Section 2**, we detail our experiment framework. In **Section 3**, we determine that while the model is invariant to the input transformation considered, several saliency methods attribute to the mean shift. In **Section 4** we discuss "reference point" methods and illustrate the importance of choosing an appropriate reference before discussing some directions of future research in **Section 5**.

## 2    THE MODEL IS INVARIANT TO A CONSTANT SHIFT IN INPUT

We show that, by construction, the bias of a neural network compensates for the mean shift resulting in two networks with identical weights and predictions. We first demonstrate this point and then describe the details of our experiment setup to evaluate the input invariance of a set of saliency methods.

We compare the attribution across two networks, $f_1(x)$ and $f_2(x)$. $f_1(x)$ is a network trained on input $x_1^i$ that denotes sample $i$ from training set $X_1$. The classification task of network 1 is:

$$f_1(x_1^i) = y^i,$$

$f_2(x)$ is a network that predicts the classification of a transformed input $x_2^i$. The relationship between $x_1^i$ and $x_2^i$ is the addition of constant vector $m_2$:

$$\forall i, x_2^i \;=\; x_1^i + m_2.$$

Network 1 and 2 differ only by construction. Consider the first layer neuron before non-linearity in $f_1(x)$:

$$z = w^T x_1 + b_1.$$

We alter the biases in the first layer neuron by adding the mean shift $\boldsymbol{m}_2$. This now becomes Network 2:

$$b_2 = b_1 - \boldsymbol{w}^T \boldsymbol{m}_2.$$

As a result the first layer activations are the same for $f_1(x)$ and $f_2(x)$:

$$z = \boldsymbol{w}^T \boldsymbol{x}_2 + b_2 = \boldsymbol{w}^T \boldsymbol{x}_1 + \boldsymbol{w}^T \boldsymbol{m}_2 + b_1 - \boldsymbol{w}^T \boldsymbol{m}_2.$$

Note that the gradient with respect to the input remains unchanged as well:

$$\frac{\partial f_1(\boldsymbol{x}_1^i)}{\partial \boldsymbol{x}_1^i} = \frac{\partial f_2(\boldsymbol{x}_2^i)}{\partial \boldsymbol{x}_2^i}.$$

We have shown that Network 2 cancels out the mean shift transformation. This means that $f_1(x)$ and $f_2(x)$ have identical weights and produce the same output for the corresponding samples, $x_1^i \in X_1$, $x_2^i \in X_2$:

$$\forall i, f_1(\boldsymbol{x}_1^i) = f_2(\boldsymbol{x}_2^i).$$

## 2.1 EXPERIMENTAL SETUP

In the implementation of this experimental framework, Network 1 is a 3 layer multi-layer perceptron with 1024 ReLu-activated neurons each. Network 1 classifies MNIST image inputs in a [0,1] encoding. Network 2 classifies MNIST image inputs in a [-1,0] MNIST encoding. The first network is trained for 10 epochs using mini-batch stochastic gradient descent (SGD). The second network is created using the approach above. The final accuracy is 98.3% for both[3].

## 3 THE (IN)SENSITIVITY OF SALIENCY METHODS TO MEAN SHIFTS

In 3.1 we introduce key approaches to the classification of inputs as salient and the saliency methods we evaluate. In 3.2 we find that gradient and signal methods are input invariant. In 3.3 we find that most attribution methods considered have points where they start to break down.

## 3.1 SALIENCY METHODS CONSIDERED

Most saliency research to date has centered on convolutional neural networks. These saliency methods broadly fall into three different categories:

1. **Gradients (Sensitivity)** (Baehrens et al., 2010; Simonyan et al., 2014)) shows how a small change to the input affects the classification score for the output of interest.

2. **Signal methods** such as DeConvNet (Zeiler & Fergus, 2014), Guided BackProp (Springenberg et al., 2015) and PatternNet (Kindermans et al., 2017) aim to isolate input patterns that stimulate neuron activation in higher layers.

3. **Attribution methods** such as Deep-Taylor Decomposition (Montavon et al., 2017) and Integrated Gradients (Sundararajan et al., 2017) assign importance to input dimensions by decomposing the value $y_j$ at an output neuron $j$ into contributions from the individual input dimensions:

   $$\boldsymbol{s}_j = A(\boldsymbol{x})_j.$$

   $\boldsymbol{s}_j$ is the decomposition into input contributions and has the same number of dimensions as $\boldsymbol{x}$, $A(\boldsymbol{x})_j$ signifies the attribution method applied to output $j$ for sample $\boldsymbol{x}$. Attribution methods are distinguished from gradients by the insistence on *completeness*: the sum of all attributions should be approximately equal to the original output $y_i$.

---

[3]Although there is a gap between this and the state of art, the gap does not significantly influence our findings.

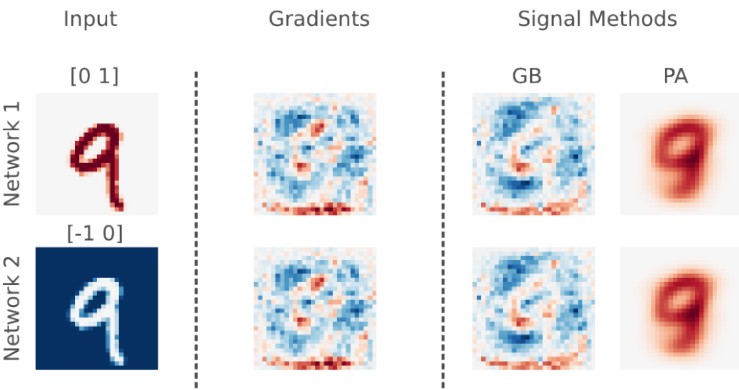

Figure 2: Evaluation of gradient and signal method sensitivity using MNIST with a [0,1] encoding for network $f_1$ and a [-1,0] encoding for network $f_2$. Both gradients and signal methods produce identical saliency heatmaps for both networks.

We consider the input invariance of each category separately (by evaluating raw gradients, Guided-Backprop, Integrated Gradients and Deep Taylor Decomposition) and also benchmark the input invariance of SmoothGrad ((Smilkov et al., 2017)), a method that wraps around an underlying saliency approach and uses the addition of noise to produce a sharper visualization of the saliency heatmap.

The experiment setup and methodology is as described in **Section 2**. Each method is evaluated by comparing the saliency heatmaps for the predictions of network 1 and 2, where $x_2^i$ is simply the mean shifted input ($x_1^i + m_2$). A saliency method that is not input invariant will not produce identical saliency heatmap for Network 1 and 2 despite the mean shift of the input.

### 3.2 THE SENSITIVITY AND SIGNAL METHODS SATISFY INPUT INVARIANT

Sensitivity and signal methods are not sensitive to the mean shift in inputs. In Fig. 2 raw gradients, PatternNet (PN, Kindermans et al. (2017)) and Guided Backprop (GB, Springenberg et al. (2015)) produce identical saliency heatmaps for both networks. Intuitively, gradient, PN and GB are input invariant given that we are comparing two networks with an identical $f(x)$. Both methods determine attribution entirely as a function of the network/pattern weights and thus will be input invariant as long as we are comparing networks with identical weights.

In the same manner, we can say that these methods will not be input invariant when comparing networks with different weights (even if we consider models with different architectures but identical predictions for every input).

### 3.3 THE ATTRIBUTION METHODS CONSIDERED

We evaluate the following attribution methods: gradient times input (GI), integrated gradients (IG, Sundararajan et al. (2017)) and the deep-taylor decomposition (DTD, Montavon et al. (2017)).

In 3.3.1 we find GI to be sensitive to meaningless input shifts. In 3.3.2 we group discussion of IG and DTD under "reference point" methods because both require that attribution is done in reference to a defined point. We find that the choice of reference point can cause input invariance to become arbitrary.

### 3.3.1 GRADIENT TIMES INPUT IS SENSITIVE TO MEAN SHIFT OF INPUTS

We find that the multiplication of raw gradients by the image breaks attribution reliability. In Fig. 3 GI produces different saliency heatmaps for both networks.

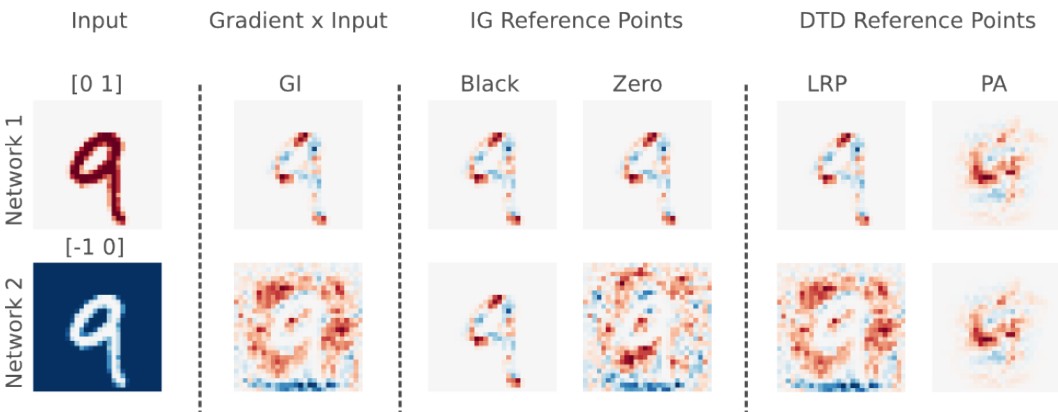

Figure 3: Evaluation of attribution method sensitivity using MNIST with a [0,1] encoding for network $f_1$ and a [-1,0] encoding for network $f_2$. Gradient x Input, IG and DTD with a zero reference point, which is equivalent to LRP (Bach et al., 2015; Montavon et al., 2017), are not reliable and produce different attribution for each network. IG with a black image reference point and DTD with a PA reference point are not sensitive to the transformation of the input.

In 3.2 we determined that a heatmap of gradients alone is not sensitive to the input transformation. GI multiplies the gradient w.r.t. the input with the input image.

$$s_j = \frac{\partial f(\boldsymbol{x})_j}{\partial \boldsymbol{x}} \odot \boldsymbol{x}.$$

Multiplying by the input means attribution is no longer reliable because the input shift is carried through to final attribution. Naive multiplication by the input, as noted by Smilkov et al. (2017), also constrains attribution without justification to inputs that are not 0.

### 3.3.2 RELIABILITY OF REFERENCE POINT METHODS DEPENDS ON THE CHOICE OF REFERENCE

Both Integrated Gradients (IG, Sundararajan et al. (2017)) and Deep Taylor Decomposition (DTD, Montavon et al. (2017)) determine the importance of inputs relative to a reference point. DTD refers to this as the root point and IG terms the reference point a baseline. The choice of reference point is not determined *a priori* by the method and instead left to end user.

The choice of reference point determines all subsequent attribution. In Fig. 1 IG and DTD show different attribution depending on the choice of reference point. We show that the certain reference point also cause IG and DTD to are not input invariant.

**Integrated gradients (IG)**    Integrated Gradients (IG, (Sundararajan et al., 2017)) attribute the predicted score to each input with respect to a baseline $\boldsymbol{x}_0$. This is achieved by constructing a set of inputs interpolating between the baseline and the input.

$$s_j = (\boldsymbol{x} - \boldsymbol{x}_0) \odot \int_{\alpha=0}^{1} \frac{\partial f(\boldsymbol{x}_0 + \alpha(\boldsymbol{x} - \boldsymbol{x}_0))_i}{\partial \boldsymbol{x}} d\alpha$$

Since this integral cannot be computed analytically, it is approximated by a finite sum ranging over $\alpha \in [0, 1]$.

$$s_j = (\boldsymbol{x} - \boldsymbol{x}_0) \odot \sum_{\alpha} \frac{\partial f(\boldsymbol{x}_0 + \alpha(\boldsymbol{x} - \boldsymbol{x}_0))}{\partial \boldsymbol{x}}.$$

We evaluate whether two possible IG reference points satisfy input invariance. Firstly, we consider an image populated uniformly with the minimum pixel from the dataset ($\boldsymbol{x}_0 = min(\boldsymbol{x})$) (black image) and a zero vector image. We find that IG attribution under certain reference points is not input invariant. In Fig. 3, IG with black reference point produces identical attribution heatmaps whereas IG with a zero vector reference point is not input invariant.

IG using a black reference point is not sensitive to the mean input shift because $\boldsymbol{x}_0 = min(\boldsymbol{x})$ is determined after the mean shift of the input so the difference between $\boldsymbol{x}$ and $\boldsymbol{x}_0$ remains the same for both networks. In network 1 this is $(\boldsymbol{x}_1) - min(\boldsymbol{x}_1)$ and in network 2 this is $(\boldsymbol{x}_2 + \boldsymbol{m}_2) - min(\boldsymbol{x}_2 + \boldsymbol{m}_2)$.

IG with a zero vector reference point is not input invariant because while the difference in network 1 is $(\boldsymbol{x}_1 - \boldsymbol{x}_0)$, the difference in network 2 becomes $(\boldsymbol{x}_2 + \boldsymbol{m}_2) - \boldsymbol{x}_0$.

**Deep Taylor Decomposition (DTD)**   determines attribution relative to a reference point neuron. DTD can satisfy input invariant if the right reference point is chosen. In the general formulation, the attribution of an input neuron $j$ is initialized to be equal to the output of that neuron. The attribution of other output neurons is set to zero. This attribution is backpropagated to its input neurons using the following distribution rule where $s_j^l$ is the attribution assigned to neuron $j$ in layer $l$:

$$s_j^{output} = y, \qquad s_{k \neq j}^{output} = 0, \qquad \boldsymbol{s}^{l-1,j} = \frac{\boldsymbol{w} \odot (\boldsymbol{x} - \boldsymbol{x}_0)}{\boldsymbol{w}^T \boldsymbol{x}} s_j^l.$$

We evaluate the input invariance of DTD using a reference point determined by Layer-wise Relevance Propagation (LRP) and PatternAttribution (PA). In Fig. 3, DTD satisfies input invariance when using a reference point defined by PA however it loses reliability when using a reference point defined by LRP.

Layer-wise Relevance Propagation (LRP, Bach et al. (2015)) is sensitive to the input shift because it is a case of DTD where a zero vector is chosen as the root point.[2]. The back-propagation rule becomes:

$$s_j^{output} = y, \qquad s_{k \neq i}^{output} = 0, \qquad \boldsymbol{s}^{l-1,j} = \frac{\boldsymbol{w} \odot \boldsymbol{x}}{\boldsymbol{w}^T \boldsymbol{x}} s_j^l.$$

$\boldsymbol{s}^{l-1,j}$ depends only upon the input and so attribution will change between network 1 and 2 because $\boldsymbol{x}_1$ and $\boldsymbol{x}_2$ differ by a constant vector.

PatternAttribution (PA) satisfies input invariance because the reference point $\boldsymbol{x}_0$ is defined as the natural direction of variation in the data (Kindermans et al., 2017). The natural direction of the data is determined based upon covariances and thus compensates explicitly for the mean in the data. Therefore it is by construction input invariant.

The PA root point is:
$$\boldsymbol{x}_0 = \boldsymbol{x} - \boldsymbol{a}\boldsymbol{w}^T\boldsymbol{x} \tag{1}$$

where $\boldsymbol{a}^T\boldsymbol{w} = 1$.

In a linear model:
$$\boldsymbol{a} = \frac{\text{cov}[\boldsymbol{x}, y]}{\boldsymbol{w}^T \text{cov}[\boldsymbol{x}, y]}. \tag{2}$$

For neurons followed by a ReLu non-linearity the vector $\boldsymbol{a}$ accounts for the non-linearity and is computed as:
$$\boldsymbol{a} = \frac{E_+[\boldsymbol{x}, y] - E_+[\boldsymbol{x}]E[y]}{\boldsymbol{w}^T(E_+[\boldsymbol{x}, y] - E_+[\boldsymbol{x}]E[y])}.$$

Here $E_+$ denotes the expectation taken over values where $y$ is positive.

---

[2]This case of DTD is called the $z - rule$ and can be shown to be equivalent to Layer-wise Relevance Propagation (Bach et al., 2015; Montavon et al., 2017). Under specific circumstances, LRP is also equivalent to the gradient times input (Kindermans et al., 2016; Shrikumar et al., 2016).

Figure 4: Smoothgrad inherits the invariance properties of the underlying attribution method. SG is not sensitive to the input transformation for gradient and signal methods (SG-PA and and SG-GB). SG lacks input invariance for integrated gradients and deep taylor decomposition when a zero vector refernce point is used, but is not sensitive when PatternAttribution (SG-PA) or a black image (SG-Black) are used. SG is not input invariant for gradient x input.

PA reduces to the following step:

$$s_i^{output} = y, \qquad s_{j \neq i}^{output} = 0, \qquad \boldsymbol{s}^{l-1,i} = \boldsymbol{w} \odot \boldsymbol{a} s_i^l.$$

The vector $\boldsymbol{a}$ depends upon covariances and thus removes the mean shift of the input. The attribution for both networks is identical.

## 3.4 SMOOTHGRAD INHERITS THE SENSITIVITY PROPERTIES OF UNDERLYING METHODS

SmoothGrad (SG, Smilkov et al. (2017)) replaces the input with N identical versions of the input with added random noise. These noisy inputs are injected into the underlying attribution method and final attribution is the average attribution across N. For example, if the underlying methods are gradients w.r.t. the input. $g(\boldsymbol{x})_j = \frac{\partial f(\boldsymbol{x})_j}{\partial \boldsymbol{x}}$ SG becomes:

$$\frac{1}{N} \sum_{i=1}^{N} g(\boldsymbol{x} + \mathcal{N}(0, \sigma^2))_j$$

SG often results in aesthetically sharper visualizations when applied to multi-layer neural networks with non-linearities. SG does not alter the attribution method itself so will always inherit the input

## Attribution Under Constant Vector Shift

Figure 5: Evaluation of attribution method sensitivity using MNIST. Gradient x Input, IG with both a black and zero reference point and DTD with a LRP reference point, do not satisfy input invariance and produce different attribution for each network. DTD with a PA reference point are not sensitive to the transformation of the input.

invariance of the underlying method. In Fig. 4 applying SG on top of gradients and signal methods (PA and GB) produces identical saliency maps. SG is not input invariant when applied to gradient x input, LRP and zero vector reference points which compares SG heatmaps generated for all methods discussed so far. SG is not sensitive to the input transformation when applied to PA and a black image.

## 4 THE IMPORTANCE OF CHOOSING AN APPROPRIATE REFERENCE POINT

IG and DTD do not satisfy input invariance under certain reference points. The reference point determines subsequent attribution. In fig.1 attribution visually diverges for the same method if multiple reference points are considered.

A reasonable reference point for IG and DTD will naturally depend upon domain and task. Unintentional misrepresentation of the model is very possible when the choice of vantage point can lead to very different results. Thus far, we have discussed attribution for image recognition tasks with the assumption that preprocessing steps are known and visual inspection of the points determined to be salient is possible. For Audio and Language based models where input interaction is more intricate, attribution becomes even more challenging.

If we cannot determine the implications of reference point choice, we are limited in our ability to say anything about the reliability of the method. To demonstrate this point, we construct a constant shift of the input that takes advantage of attribution points of failure discussed thus far. Almost all methods are sensitive to this input transformation which results in a misleading explanation of the model prediction.

Network 1 is the same as introduced in **Section 2**. We consider a transformation $x_2^i$ of the input $x_1^i$ which is a MNIST image.

In this experiment $m_2$ is the vector of an image of a checkered box. Consistent with **Section 2** the relationship between $x_1^i$ and the transformed input $x_2^i$ is the addition of a constant vectors $m_2$.

$$\forall i, x_2^i \;=\; x_1^i + m_2.$$

Network 2 is identical to network 1 by construction (see **Section 2**). Note that $x_2^i$ and $x_3^i$ are scored by Network 2 separately.

"Cat"astrophic Attribution Failure

MNIST + Constant Shift

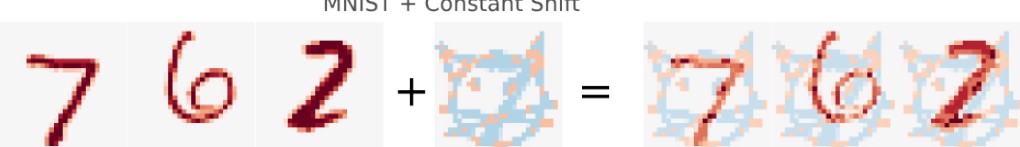

Attribution Methods

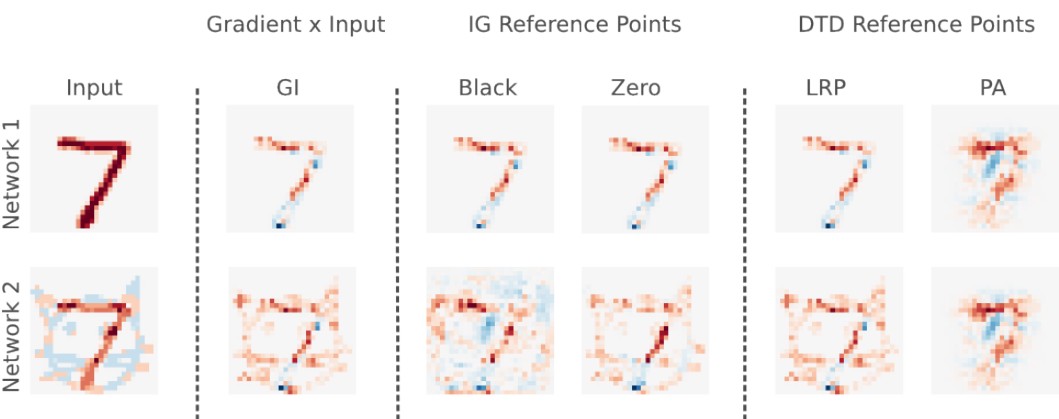

Figure 6: Evaluation of attribution method sensitivity using MNIST. Gradient x Input, IG with both a black and zero reference point and DTD with a LRP reference point, do not satisfy input invariance and produce different attribution for each network. DTD with a PA reference point are not sensitive to the transformation of the input.

In Fig. 5 all attribution methods except for PA are sensitive to this constant shift. The result is that we are able to manipulate the attribution heatmap of an MNIST prediction so that the chosen sample $\hat{x}$ appears. Using a black image as a reference point for IG no longer satisfies input invariance (as it did in the experiments in **Section 3**).

The sample $\hat{x}$ can be any abitrary vector. We conduct the same experiment with a hand drawn kitten image. We construct $m_2$ by choosing a desired attribution $\hat{s}$ that should be assigned to a specific sample $\hat{x}$ when the gradient is multiplied with the input.

We compute a $m_2$ that will ensure the specific $x_2^i$ receives the desired attribution as follows:

$$m_2 = \frac{\hat{s}}{\frac{\partial f_1(x)}{\partial x}} - x.$$

To make sure that the original image is still recognizable as belonging to its class, we clip the shift to be within [-.3,.3]. Of course, the attributions of the other samples in the dataset is impacted too.

In Fig. 6 we see that we are again able to purposefully misrepresent the explanation of the model prediction.

It is important to note that that some of these methods would have satisfied input invariance if the data had been normalized prior to attribution. For example, IG with a black baseline will satisfy input invariance if the data is always normalized. However, this is far from a systematic treatment of the reference point selection and there are cases outside of our experiment scope where this would not be sufficient. We believe an open research question is furthering the understanding of reference point choice that guarantee reliability without relying on case-by-case solutions.

## 5 CONCLUSION

Saliency methods are powerful tools to gain intuition about our model. We consider some examples that can cause a break in the reliability of these methods. We show that we are able to purposefully create a deceptive explanation of the network using a hand drawn kitten image.

We introduce *input invariance* as a prerequisite for reliable attribution. Our treatment of input invariance is restricted to demonstrating there is at least one input transformation that causes attribution to fail. We hope this work drives further discussion on this subject. We also acknowledge that saliency methods may still provide intuition for image recognition tasks even if they are not input invariant. Our work is motivated in part because while we can visually inspect for catastrophic attribution failure in images, other modalities (like audio or word vectors) are more opaque and prone to unintentional misrepresentation.

### ACKNOWLEDGEMENTS

We will thank everyone in the final version. But it goes without saying that we are grateful to our parents.

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
