# OpenReview forum: "The (Un)reliability of saliency methods"
_ICLR.cc/2018/Conference — Reject_

### Official Review · AnonReviewer1 · 2017-11-27
**The (Un)reliability of saliency methods**

**Rating:** 5
**Confidence:** 3

**Review:**

The scope of the paper is interesting i.e. taking a closer look at saliency methods in view of explaining deep learning neural networks. The authors state that saliency methods that do not satisfy an input invariance property can be misleading.

On the other hand the paper can be improved in my opinion in different aspects:
- it would be good to be more precise on the type of invariance (e.g. translation invariance, rotation invariance etc.) or is the paper only about invariance to mean shifts? I suggest to explain in the introduction which type of invariances have been considered in the area of deep learning and then position the paper relative to it.
- in the introduction the authors talk about an "invariance axiom": it was difficult to see where in the paper this axiom is precisely stated.
- While in section 2.1 specifies a 3-layer MLP as the considered deep learning network. It is not clear why CNN haven't been used here (especially because the examples are on MNIST images), while in section 3.1 this is mentioned.
- I think that the conclusion with respect to invariances could also depend on the choice of the activation function. Therefore the authors should from the beginning make more clear to which class of deep learning networks the study and conclusions apply.
- From section 3.1 it becomes rather unclear which parts of the paper relate to the literature and which parts relate to section 2.1. Also the new findings or recommendations are not clear.

---

> ### Author Response · Authors · 2018-01-05
> **Response**
>
> Quote:
> “- it would be good to be more precise on the type of invariance (e.g. translation invariance, rotation invariance etc.) or is the paper only about invariance to mean shifts? I suggest to explain in the introduction which type of invariances have been considered in the area of deep learning and then position the paper relative to it.”
>
> Answer:
> We agree with the reviewer that we could have been more explicit. Our broad motivation is determining failure points by formulating unit tests of commonly used saliency methods. We propose that one criteria methods should fulfill in order to be reliable is input invariance (II). We benchmark methods by considering one possible transformation, a mean shift of the input. The reviewer is correct that additional transformation to the inputs should also be considered but that is beyond the current scope.
>
>
>
> Quote:
>  “- in the introduction the authors talk about an "invariance axiom": it was difficult to see where in the paper this axiom is precisely stated.”
>
> Answer:
> The term axiom was used in the same vein as prior work (integrated gradients) to articulate a desirable property. On reflection upon the reviewers feedback we agree that there may be more suitable terminology. A rephrasing consistent with our original intent is that input invariance is a desideratum of interpretability methods.
>
>
>
> Quote:
>  “- While in section 2.1 specifies a 3-layer MLP as the considered deep learning network. It is not clear why CNN haven't been used here (especially because the examples are on MNIST images), while in section 3.1 this is mentioned. “
>
>
> Answer:
> We argue attribution methods should work for all architectures. Therefore the validity of a test does not depend on the architecture chosen. This also implies that a test might not be possible for all architectures.
>
>
>
> Quote:
> “- I think that the conclusion with respect to invariances could also depend on the choice of the activation function. Therefore the authors should from the beginning make more clear to which class of deep learning networks the study and conclusions apply.”
>
> Answer:
> The given test does not depend on the activation function since from the first linear operation all activations are identical. However, we do agree with the reviewer that in future, more advanced tests, the activation function could play a role.
>
>
>
> Quote:
>  “- From section 3.1 it becomes rather unclear which parts of the paper relate to the literature and which parts relate to section 2.1. Also the new findings or recommendations are not clear.”
>
> Answer:
> We acknowledge the reviewer style feedback and agree that we could be more clear. To clarify, the key contributions of this manuscript are, which we will add to a future version of  the manuscript:
> - determine whether commonly used saliency methods reliably attribute by considering the  input invariance of said methods.
> -  Our first recommendation is that new methods be benchmarked against this test.
> - Our second recommendation is that this serves as a starting point for considering additional failure points. By extending our repertoire of tests we can develop more robust interpretability methods. This is crucial for an emerging field where the lack of ground truth means there there is no known way of measuring success. If we can reliably determine failure cases and fix these, we can weigh how to use these methods going forward.

---

### Official Review · AnonReviewer2 · 2017-11-27
**Interesting idea but missing good motiviation and dicussion**

**Rating:** 4
**Confidence:** 4

**Review:**

The authors explore how different methods of visualizing network decisions (saliency methods) react to mean shifts of the input data by comparing them on two networks that are build to compensate for this mean shift. With the emergence of more and more saliency methods, the authors contribute an interesting idea to a very important and relevant discussion.

However, I'm missing a more general and principled discussion. The question that the authors address is how different saliency methods react to transformations of the input data. Since the authors make sure that their two models compensate for these transformation, the difference in saliency can be only due to underlying assumptions about the input data made by the saliency methods and therefore the discussion boils down to which invariance properties are justified for which kind of input -- it is not by chance that the attribution methods that work are exactly those that extract statistics from the input data and therefore compensate for the input transformation: IG with black reference point and Pattern Attribution.
The mean shift explored by the authors assumes that there is no special point in the input space (especially that zero is not a special point).
However, since images usally are considered bounded by 0 and 1 (or 255), there are in fact two special points (as a side note, in Figure 2 left column, the two inputs look very different which might be due to the fact that it is not at all obvious how to visualize "image" input that does not adhere to the common image input structure).
Would the authors argue that scaling the input with a positive factor should also lead to invariant saliency methods?
What about scaling with a negative factor?
I would argue that if the input has a certain structure, then it should be allowed for the saliency method to make use of this structure.

Minor points:

Understanding the two models in section 3 is a bit hard since the main point (both networks share the weights and biases except for the bias of the first layer) is only said in 2.1

---

> ### Author Response · Authors · 2018-01-05
> **Response**
>
> Quote:
> “However, I'm missing a more general and principled discussion. The question that the authors address is how different saliency methods react to transformations of the input data. Since the authors make sure that their two models compensate for these transformation, the difference in saliency can be only due to underlying assumptions about the input data made by the saliency methods and therefore the discussion boils down to which invariance properties are justified for which kind of input -- it is not by chance that the attribution methods that work are exactly those that extract statistics from the input data and therefore compensate for the input transformation: IG with black reference point and Pattern Attribution.”
>
> Answer:
> The reviewer is correct in stating that we designed the experiment in such a way that the transformation does not affect the model predict or weights.This allows for a principled evaluation of input invariance which is the contribution of this manuscript.
>
>
>
>
> Quote:
> “The mean shift explored by the authors assumes that there is no special point in the input space (especially that zero is not a special point).
> However, since images usually are considered bounded by 0 and 1 (or 255), there are in fact two special points (as a side note, in Figure 2 left column, the two inputs look very different which might be due to the fact that it is not at all obvious how to visualize "image" input that does not adhere to the common image input structure).”
>
> Answer:
> Arguably, even zero is not a special point, since mean shifts of the data can be compensated for by the biases of the first layer in the network. The encoding of 0 to 1 is common but not necessarily special, furthermore inception using a encoding of [-1,1] and resnet uses 0 mean. Therefore it is unclear whether a reference point of  0 or 1 given a [0,1] encoding is more special than using the mean of the image. The question of what is a good reference (i.e. special point)  is relevant and an open research question for both Deep Taylor Decomposition and Integrated Gradients.
> For the Deep-Taylor decomposition, PatternAttribution proposes a learned reference point that is invariant to the mean vector shift since it is based on covariances. It is not yet clear how to do this for IG, and we encourage the community to solve this open problem.
>
>
>
>
> Quote:
> “Would the authors argue that scaling the input with a positive factor should also lead to invariant saliency methods?
> What about scaling with a negative factor?
> I would argue that if the input has a certain structure, then it should be allowed for the saliency method to make use of this structure.”
>
> Answer:
> If we scale the image with a positive factor, the weights would be required to compensate for this change to ensure the activations remain the same. The same holds for scaling with a negative factor. For this reason, all attribution methods would (not the signal or gradients) remain intact.
> The problem is not that the input has a specific structure and that the saliency method picks up on this structure. The issue is that we included a mean shift, which the network compensates for effectively. This mean shift does not contain class information, yet it dominates the attribution.

---

### Official Review · AnonReviewer3 · 2017-12-02
**Poorly motivated invariance property**

**Rating:** 4
**Confidence:** 4

**Review:**

Saliency methods are effective tools for interpreting the computation performed by DNNs, but evaluating the quality of interpretations given by saliency methods are often largely heuristic. Previous work has tried to address this shortcoming by proposing that saliency methods should satisfy "implementation invariance", which says that models that compute the same function should be assigned the same interpretation. This paper builds on this work by proposing and studying "input invariance", a specific kind of implementation invariance between two DNNs that compute identical functions but where the input is preprocessed in different ways. Then, they examine whether a number of existing saliency methods satisfy this property.

The property of "implementation invariance" proposed in prior work seems poorly motivated, since the entire point of interpretations is that they should explain the computation performed by a specific network. Even if two DNNs compute the same function, they may do so using very different computations, in which case it seems natural that their interpretations should be different. Nevertheless, I can believe that the narrower property of input invariance should hold for saliency methods.

A much more important concern I have is that the proposed input invariance property is not well motivated. A standard preprocessing step for DNNs is to normalize the training data, for example, by subtracting the mean and dividing by the standard deviation. Similarly, for image data, pixel values are typically normalized to [0,1]. Assuming inputs are transformed in this way, the input invariance property (for mean shift) is always trivially satisfied. The paper does not justify why we should consider networks where the training data is not normalized is such a way.

Even if the input is not normalized, the failures they find in existing saliency methods are typically rather trivial. For example, for the gradient times input method, they are simply noting that the interpretation is translated by the gradient times the mean shift. The paper does not discuss why this shift matters. It is not at all clear to me that the quality of the interpretation is adversely affected by these shifts.

I believe the notion that saliency methods should be invariant to input transformations may be promising, but more interesting transformations must be considered -- as far as I can tell, the property of invariance to linear transformations to the input does not provide any interesting insight into the correctness of saliency methods.

---

> ### Author Response · Authors · 2018-01-05
> **Response**
>
> Quote:
>  “A much more important concern I have is that the proposed input invariance property is not well motivated. A standard preprocessing step for DNNs is to normalize the training data, for example, by subtracting the mean and dividing by the standard deviation. Similarly, for image data, pixel values are typically normalized to [0,1].”
>
> Answer:
> In response to the statement that input invariance is not well motivated:
> Our stance is that interpretability methods should pass our test regardless of the pre-processing of the data. The onus should not be that the researcher has to ensure correct preprocess to guarantee a reliable explanation.
> We argue that even among popular architectures like resnet and inception there is no standard image normalization procedure (e.g., some are [0,1] whereas others are [-1,1]).
>
>
>
> Quote:
> “Even if the input is not normalized, the failures they find in existing saliency methods are typically rather trivial. For example, for the gradient times input method, they are simply noting that the interpretation is translated by the gradient times the mean shift. The paper does not discuss why this shift matters. It is not at all clear to me that the quality of the interpretation is adversely affected by these shifts.”
>
> Answer:
> The failures are regarded by the reviewer as unsubstantial given that the explanation is still interpretable. We acknowledge that we choose a simple transformation to illustrate a simple point of failure. This is sufficient to show that a point of failure exists. Furthermore, the motivation for interpretability research is to explain model predictions for data we do not yet fully understanding. Illustrating that many methods fail in the simple case is therefore valuable.
>
>
>
> Quote:
> “I believe the notion that saliency methods should be invariant to input transformations may be promising, but more interesting transformations must be considered -- as far as I can tell, the property of invariance to linear transformations to the input does not provide any interesting insight into the correctness of saliency methods.”
>
> Answer:
> We agree with the reviewer that future work on additional input transformations is needed. However, we justify our approach by the logic that a single transformation is sufficient to demonstrate a method is unreliable. Our contribution is to formulate a unit test that can detect a specific failure point and allows us to proactively improve existing methods. Note that our unit test does not account for all possible failure points, and thus more research is needed to consider whether the methods deemed to be reliable remain so. We invite other researchers to design additional unit tests.

---

### Author Response · Authors · 2018-01-05
**General response**

Before replying to the individual comments let us restate the motivation of this work.
The key motivation is to raise awareness about potential issues with methods for interpretability. This is a young field and developing tests for reliable methods can help our field to become (even) more rigorous and scientific.

Our work is based on the following observations:

- Interpretability methods should be reliable. We define reliability as being insensitive to factors that do not affect the decision making process learnt by the model.

- The utility of these methods, particularly in sensitive domains like health care, depends upon demonstrating reliability regardless of the model architecture and data preprocessing chosen.

- There is no ground truth for what a model finds important which has led to a large number of methods with surprisingly different outcomes. Benchmarking  the “quality” of saliency methods is a difficult and unsolved problem.

- Our framework for measuring the utility of saliency methods is to determine points of failure in reliability. A single point of failure is sufficient to show that a method is unreliable; an analogue to proving by counterexample.

- Determining failure points allows us to proactively improve existing methods. We invite other researchers to find additional failure points. Determining where methods fall short is a crucial step in choosing appropriate methods for given tasks and improving these methods.


Based on the points above we argue that our contribution is important because we demonstrate that a simple, commonly used transformation, causes many (recently published) saliency methods to fail. It is necessary to initiate this conversation because visually determining points of failure is far from trivial in high dimensional data and in modalities other than images such as audio and word vectors.

Proving that many methods are unreliable using a very simple transformation case is a starting point for the community to develop more reliable methods. It is akin to a unit test which does not guarantee that your code solves the correct problem but highlights when your code clearly does not solve the problem. In this paper we formulate a single “unit test” allows us to identify points of failure and develop robust methods in the future.

---

### Decision · Program_Chairs · 2018-01-29
**ICLR 2018 Conference Acceptance Decision**

**Decision:**

Reject

**Comment:**

This paper showcases how saliency methods are brittle and cannot be trusted to obtain robust explanations. They define a property called input invariance that they claim all reliable explanation methods must possess. The reviewers have concerns regarding the motivation of this property in terms of why is it needed. This is not clear from the exposition. Moreover, even after having the opportunity to update the manuscript they seem to have not touched upon this issue other than providing a generic response.